# Predictive Factors Indicative of Hemithyroidectomy and Close Follow-Up versus Bilateral Total Thyroidectomy for Aggressive Variants of Papillary Thyroid Cancer

**DOI:** 10.3390/cancers14112757

**Published:** 2022-06-02

**Authors:** In A Lee, Gilseong Moon, Seokmin Kang, Kang Hee Lee, Sun Min Lee, Jin Kyong Kim, Cho Rok Lee, Sang-Wook Kang, Jong Ju Jeong, Kee-Hyun Nam, Woong Youn Chung

**Affiliations:** 1Department of Surgery, College of Medicine, Severance Hospital, Yonsei University, Seoul 03722, Korea; anzelina@yuhs.ac (I.A.L.); gsmoon@yuhs.ac (G.M.); junbumin@yuhs.ac (S.K.); leekh8695@yuhs.ac (K.H.L.); ssun289@yuhs.ac (S.M.L.); jkkim3986@yuhs.ac (J.K.K.); oralvanco@yuhs.ac (S.-W.K.); khnam@yuhs.ac (K.-H.N.); woungyounc@yuhs.ac (W.Y.C.); 2Department of Surgery, Yongin Severance Hospital, Yongin-si 16995, Korea; crlee@yuhs.ac

**Keywords:** aggressive variants of papillary thyroid cancer, hemithyroidectomy, follow-up, thyroidectomy, hemithyroidectomy

## Abstract

**Simple Summary:**

This study evaluated the need for additional surgical treatment in patients diagnosed with aggressive variants of papillary thyroid cancer after lobectomy. With the increase in the frequency of early diagnosis, the detection of papillary thyroid cancer pathologically belonging to the aggressive variant is also increasing. Therefore, there is growing concern regarding the aggressive treatment of encapsulated aggressive variants of papillary thyroid cancer without invasive features. We suggest that close follow-up can be performed without any additional surgical treatment in patients with low-risk aggressive variants of papillary thyroid cancer incidentally detected after hemithyroidectomy. The patients should be provided sufficient consultation with clinicians.

**Abstract:**

The diagnostic and treatment rates of early thyroid cancer have been increasing, including those of aggressive variants of papillary thyroid cancer (AVPTC). This study aimed to analyze the need for completion total thyroidectomy after lobectomy for clinically low-to-intermediate-risk AVPTC. Overall, 249 patients who underwent hemithyroidectomy (HT, n = 46) or bilateral total thyroidectomy (BTT, n = 203) for AVPTC between November 2005 and December 2019 at our single institution were examined. The average follow-up period was 14.9 years, with a recurrence rate of 4.3% and 10.8% in the HT and BTT groups, respectively. Multivariate Cox analysis revealed that palpable tumor on the neck during evaluation (HR, 2.7; 95% CI, 1.1–6.4; *p* = 0.025), clinical N1b (HR, 8.3; 95% CI, 1.1–63.4; *p* = 0.041), tumor size (cm) (HR, 1.3; 95% CI, 1.0–1.7; *p* = 0.036), gross extrathyroidal extension (HR, 3.1; 95% CI, 1.4–7.0; *p* = 0.007), and pathologic T3b (HR, 3.4; 95% CI, 1.0–11.4; *p* = 0.045) or T4a (HR, 6.0; 95% CI, 1.9–18.8; *p* = 0.002) were associated with an increased risk of recurrence. Incidentalomas identified during diagnosis had a significantly lower risk of recurrence (HR, 0.4; 95% CI, 0.2–0.9; *p* = 0.033). Close follow-up may be performed without completion total thyroidectomy for AVPTC found incidentally after HT.

## 1. Introduction

Papillary thyroid cancer (PTC) is the most common endocrine cancer, and its incidence increases annually by 6.5% [1]. Despite good prognosis, with a 10-year survival rate of approximately 93%, there are subtypes of aggressive variants of PTC (AVPTC) that are heterogeneous and have clinicopathologic and molecular features that are distinct from those of classic PTC (cPTC) [2]. The American Thyroid Association (ATA) guidelines classify the tall cell variant (TCV), columnar cell variant (CCV), hobnail variant (HV), diffuse sclerosing variant (DSV), and solid variant (SV) as being of intermediate risk [3]. Compared with cPTC, TCV is associated with higher extrathyroidal extension, more extensive locoregional or distant metastasis, and poorer prognosis and is also an independent factor for an aggressive disease course [4,5,6,7,8]. CCV and HV have a high risk of distant metastasis and cancer-specific mortality [9,10,11]. These three subtypes/variants are often accompanied by BRAF (V600E) mutation [6,12]. Compared with cPTC, DSV is more prevalent in young female patients with radiation exposure, is more often diagnosed at an advanced stage, and has a higher risk of recurrence and distant metastasis. However, the prognosis of DSV is not poor because of its good response to treatment [13,14,15,16,17]. Meanwhile, SV, distinguished from poorly differentiated thyroid carcinoma, is associated with a higher rate of distant metastasis and slightly less favorable prognosis [18,19]. The DSV and SV subtypes are accompanied by RET/PTC rearrangements and BRAF mutation [15,18].

The recently revised 2015 ATA guidelines state that even the intermediate-to-slightly high-risk group of patients without gross extrathyroidal extension (ETE) can undergo lobectomy, considering the patient’s status and preferences [3,20]. As such, there is an increasing tendency of surgeons to pursue conservative management in the surgical treatment of PTC. Therefore, the necessity to complete total thyroidectomy for AVPTC, which pathologically has an intermediate risk according to the ATA guidelines, is under debate [3]. Therefore, this study aimed to analyze the need for completing a total thyroidectomy after lobectomy for AVPTC patients with a clinically low-to-intermediate risk.

## 2. Methods

### 2.1. Study Design and Patients

This retrospective study examined 249 patients who underwent hemithyroidectomy (HT, n = 46) or bilateral total thyroidectomy (BTT, n = 203) for AVPTC at Severance Hospital, Yonsei University College of Medicine, Seoul, Korea between November 2005 and December 2019. The exclusion criteria were as follows: (1) diagnosis with anaplastic thyroid cancer and poorly differentiated thyroid cancer based on the postoperative pathology results and (2) reoperation for completion total thyroidectomy after confirmation of the final pathology results. All the patients were preoperatively diagnosed via physical examination, ultrasonography (US), and neck and chest computed tomography (CT), and the preoperative malignancy classification was confirmed by US-guided fine-needle aspiration (FNA). Clinical classification of cancer and lymph nodes was performed according to the Korean Thyroid Imaging Reporting and Data System by radiologists at our institution [21]. The scope of surgery (HT or BTT) was decided in accordance with the ATA guidelines, and in some cases, conservative surgery (HT) was performed, considering the patient’s preference [3]. During the postoperative outpatient visit, the patients who underwent HT were educated about the risk of AVPTC for informed decision making about completion total thyroidectomy or active close follow-up. Clinicopathological data were collected retrospectively and stored in a dedicated database for analysis.

### 2.2. Postoperative Follow-Up

We analyzed the following clinical parameters: patient characteristics; clinical imaging findings; operative variables; extent of surgery; pathological findings; and postoperative outcomes, including the recurrence rate. For patients showing signs of recurrence on postoperative imaging, FNA was used to confirm the recurrence. Pathological examinations included assessments of cancer type, tumor size and number, gross ETE, tumor-node-metastasis (TNM) stage, number of lymph nodes (LNs) harvested, number of metastatic LNs, and BRAF and TERT promoter mutations. All the patients were followed up in the same manner, which included clinical examinations within 1 week of discharge and a 3-to-6-month follow-up comprising physical examination, neck US or CT, an assay of tumor markers (serum thyroglobulin and thyroglobulin antibody concentration), and radioactive iodine (RAI) treatment, as needed.

### 2.3. Statistical Analysis

Continuous quantitative data are presented as mean ± standard deviation, whereas categorical qualitative data are presented as percentages. Between-group comparisons were performed using the Chi-squared test or Mann–Whitney U test, as appropriate. Survival rates were examined using the Kaplan–Meier method and compared using the log-rank test. Perioperative clinicopathological findings were analyzed using univariate and multivariate Cox regression analyses to identify predictive factors of recurrence. Factors that were significant (*p* < 0.05) in the univariate analysis were included in the multivariate analysis. Contal and O’Quigley’s method and Cox proportional hazards regression analysis were used to analyze optimal cut-off of tumor size for recurrence [22]. R version 3.6.0 (The R Foundation for Statistical Computing, Vienna, Austria) was used to analyze the tumor size cut-off, and all other statistical analyses were performed using the Statistical Package for the Social Science software for Windows version 26.0 (IBM Corp., Armonk, NY, USA). *p* values < 0.05 were considered statistically significant.

## 3. Results

### 3.1. Patient Characteristics

Among 249 patients (65 male patients), 46 (10 male patients; sex ratio 1:3.6) underwent HT and 203 (55 male; sex ratio 1:2.7) underwent BTT, with no significant between-group difference noted in the proportion of female patients (*p* = 0.455). The mean age and history of hypothyroidism were similar between the HT and BTT groups (*p* = 0.195 and *p* = 0.282, respectively). Incidentaloma was the most common reason for patient visits in both the HT and BTT groups, with a significant between-group difference noted in the rates of incidentaloma (93.5% vs. 74.9%, *p* = 0.006). Patients in the HT group were mostly diagnosed with cancer due to an incidentaloma (93.5%). Meanwhile, patients in the BTT group presented with many symptoms in the advanced stage, such as anterior neck mass (18.7%), lateral neck mass (3.0%), hoarseness (2.5%), and others (dyspnea, weight loss) (1.0%), in addition to incidentaloma (74.9%). There were significant differences in the family history related to thyroid disease (thyroid cancer, other organ cancers, and hypothyroidism) between the HT and BTT groups (26.1% vs. 13.3%, *p* = 0.031). No patient had a history of radiation exposure in both the groups. The patient characteristics by group are presented in Table 1.

Table 2 shows the between-group comparison of preoperative imaging findings (US and CT) for clinical TNM and US-guided detection of tumor aggressiveness. Most patients in the HT group had a T1 stage (67.4%) and N0 stage (89.1%) disease. Meanwhile, the BTT group had significantly more advanced T and N stage compared with the HT group (*p* < 0.001 and *p* < 0.001, respectively). The majority of patients in the BTT group had T3 stage (61.1%) and N1b stage (67.0%) tumors. Moreover, the rate of metastasis at the time of diagnosis was higher in the BTT group than in the HT group [ (0%) vs. 7 (3.4%), *p* = 0.201]. The FNA cytology results for the nodule were 2 (4.3%), 1 (2.2%), 5 (10.9%), 21 (45.7%), and 17 (37.0%) for nondiagnostic, benign, atypia/follicular lesions of undetermined significance, suspicious for malignancy, and malignant, respectively, in the HT group. However, in the BTT group, there were only 15 (7.4%), 142 (70.0%), and 46 (22.7%) for atypia/follicular lesions of undetermined significance, suspicious for malignancy, and malignant, respectively, by significant difference (*p* < 0.001). Aggressive variants were found on US in 4.3% and 37.4% of the patients in the HT and BTT groups, respectively, with a significant difference (*p* < 0.001).

### 3.2. Pathological Findings and Surgical Outcomes

The pathological findings along with gene mutations are shown in Table 3. All intermediate-risk pathological subtypes of PTC were identified in the study population, with a significant difference in their distribution between the HT and BTT groups (*p* < 0.001). In the HT group, 19 (41.3%), 19 (41.3%), 6 (13.0%), 1 (2.2%), and 1 (2.2%) patient had DSV, SV, TCV, HV, and CCV, respectively. In the BTT group, 163 (80.3%), 23 (11.3%), 13 (6.4%), 3 (1.5%), and 1 (0.5%) patient had DSV, SV, TCV, HV, and CCV, respectively. The mean tumor size was 1.2 ± 0.8 cm (range, 0.4–3.5 cm) and 1.9 ± 1.2 cm (range, 0.3–7.0 cm) in the HT and BTT groups, respectively, with no significant difference (*p* = 0.226). Meanwhile, there were significant between-group differences in multiplicity and bilaterality of thyroid cancer between the groups. The multiplicity of thyroid cancer was higher in the BTT group than in the HT group (55.7% vs. 23.9%, *p* < 0.001). In total, 123 patients (60.0%) in the BTT group had bilateral thyroid cancer; of them, 36.0% had bilateral AVPTCs and 24.6% had cPTC and AVPTC. In the HT group, a clinically benign nodule on the lateral side of the thyroid cancer was identified as cPTC in 3 (6.5%) patients who underwent HT with partial thyroidectomy.

For positive metastatic LNs, the rate of LN involvement was higher in the BTT group than in the HT group (86.2% vs. 41.3%, *p* < 0.001). Further, although there was no significant between-group difference in the average number of harvested LNs (*p* = 0.091), the number of harvested metastatic LNs was significantly higher in the BTT group (6.7 ± 5.9 vs. 1.6 ± 2.4, *p* < 0.001). Additionally, more patients in the BTT group showed positive findings for perinodal soft tissue extension (53.2% vs. 2.2%, *p* < 0.001), microscopic ETE (76.8% vs. 39.1%, *p* < 0.001), and gross ETE (23.2% vs. 0.0%, *p* < 0.001). Meanwhile, there was no significant between-group difference in lymphovascular invasion (LVI) (*p* = 0.074) and positive BRAF mutation (*p* = 0.605). For positive TERT promoter mutation, all 46 patients in the HT group showed negative results, whereas 4 (16.0%) of the 25 patients in the BTT group had a TERT promoter mutation (*p* = 0.005). All patients in the HT group underwent genetic testing, but some patients in the BTT group were unable to undergo testing because of old age or loss of specimen. The BRAF and TERT promoter mutation rates were 38.9% and 8.1% for DSV, 36.4% and 0.0% for SV, 94.1% and 9.1% for TCV, 50% for HV, and 0.0% and 0.0% for CCV, respectively. In the final pathologic results, patients in the HT group had T1 (84.8%) or T2 (15.2%) stage disease and 58.7% and 41.3% had N0 and N1a stage disease, respectively. Meanwhile, for patients in the BTT group, 69.5% had T1–T2 stage disease, 30.5% had T3–T4 stage disease, and 18.7% and 67.0% had N1a and N1b stage disease, respectively.

Among patients in the BTT group, 94.6% underwent postoperative adjuvant RAI treatment. During the average follow-up period of 14.9 years, 2 patients (4.3%) in the HT group and 22 patients (10.8%) in the BTT group developed recurrence (*p* = 0.178). The two patients in the HT group had locoregional recurrence in the lateral neck node 5 years and 3.3 years after the first surgery. Meanwhile, among these 22 patients in the BTT group, 11 (5.4%) patients had locoregional recurrence, 7 (3.4%) had distant recurrence, and 4 (2.0%) had both locoregional and distant recurrence. Among the 22 patients (10.8%) in the BTT group who developed recurrence within an average of 3.4 years after the first surgery, 17 (8.4%) patients were diagnosed with DSV; 3 (1.5%) patients, SV; and 2 (1.0%) patients, TCV. Among the 17 patients diagnosed with DSV who showed recurrence, 6 (35.3%) had distant metastasis, 4 (23.5%) had lateral neck node metastasis, 4 (23.5%) had operative bed and lateral neck node metastasis, 2 (11.8%) had distant and lateral neck node metastasis, and 1 (5.9%) had operative bed metastasis. Among the three SV patients, one patient each had lateral neck node metastasis, distant metastasis, and distant and lateral neck node metastasis. Among the two TCV patients, one patient had lateral neck node metastasis and the other had distant and lateral neck node metastasis. There was no significant difference in the distribution of recurrence sites between the two groups (*p* = 0.397). The most common treatment for recurrence was reoperation with postoperative high-dose RAI in both the HT and BTT groups (Table 4).

The average, median, and interquartile ranges of follow-up for the HT and BTT groups were 55.1 ± 39.0 months (range, 12–147 months), 42.5 months, and 54 months, and 77.0 ± 44.9 months (range, 11–179 months), 73.0 months, and 79 months, respectively. There was no significant difference in disease-free survival between the groups (HT: 137.5 ± 6.4 months vs. BTT: 158.4 ± 4.1 months; log-rank *p* = 0.343; Figure 1). One patient in the BTT group died during the follow-up period because of an underlying condition.

### 3.3. Predictive Risk Factors of Recurrence

Table 5 presents the results of the univariate and multivariate Cox regression analyses of predictive risk factors of recurrence. The risk of recurrence was 0.4 times lower for incidentalomas [hazard ratio (HR), 0.4; 95% confidence interval (CI), 0.2–0.9; *p* = 0.033] and 2.7 times higher when cancer was diagnosed after a palpable neck mass was found (HR, 2.7; 95% CI, 1.1–6.4; *p* = 0.025). The risk of recurrence was 8.3 times higher in patients with clinical N1b stage on preoperative US (HR, 8.3; 95% CI, 1.1–63.4; *p* = 0.041). In addition, tumor size (HR, 1.3; 95% CI, 1.1–1.7; *p* = 0.036), gross ETE positivity (HR, 3.1; 95% CI, 1.4–7.0; *p* = 0.007), and pathology stage T3b (HR, 3.4; 95% CI, 1.0–11.4; *p* = 0.045) or T4a (HR, 6.0; 95% CI, 1.9–18.8; *p* = 0.002) increased the risk of recurrence. Conversely, age, sex, family history, clinical T stage, US-guided detection of aggressiveness, operation type, subtypes of AVPTC, perinodal soft tissue extension, microscopic ETE, multiplicity, bilaterality, LVI, LN metastasis, pathologic N stage, and postoperative adjuvant treatment methods did not have a significant impact on the risk of recurrence (Table 5).

Patients in the HT group had none or one to two risk factors, including microscopic ETE, LVI, ≥4 LNs with extranodal extension, and ≥6 LNs. The patients were further divided into two groups: with no/low-risk factor and with intermediate or high-risk factor according to revised ATA guidelines [3]. There were no significant differences in sex (*p* = 0.190), age (*p* = 0.480), family history (*p* = 0.153), extranodal extension (*p* = 0.089), bilaterality (*p* = 0.768), tumor size (*p* = 0.251), pathologic T stage (*p* = 0.088), maximum diameter of LN (*p* = 0.502), and BRAF mutation (*p* = 0.129). Meanwhile, significant differences were noted in subtypes (*p* = 0.002), multiplicity (*p* < 0.001), and clinical N1a (*p* < 0.001). For the subgroup with no risk factors, DSV was noted in 8 patients, SV in 18 patients, TCV in 6 patients, CCV in 1 patient, and HV in 1 patient. Meanwhile, for the subgroup with one to two risk factors, DSV was noted in 11 patients and SV in 1 patient.

The optimal cut-off point for tumor size predictive of recurrence was 1.4 cm (95% CI, 1.1–1.8, *p* = 0.006) [22]. The risk of recurrence was 3.3 times higher if tumor size was ≥1.4 cm (Harrell’s c-index, 0.666; Table 6) [23].

## 4. Discussion

The worldwide incidence of thyroid cancer, predominantly PTC, has steadily increased over recent decades, but the cancer-specific mortality rate of thyroid cancer has declined [24,25]. The trends also show an increase in the diagnosis of PTC owing to the development of diagnostic techniques [26,27]. A study by Hadiza et al. using the Surveillance, Epidemiology, and End Results database (1988–2008) reported that the incidence of cPTC increased by 60.8%, whereas that of DSV and TCV of cancer, including AVPTC, increased by 126% (*p* for trend = 0.052) and 158% (*p* for trend = 0.002), respectively [5]. This study sought to evaluate the need for additional surgical treatment for patients diagnosed with AVPTC after lobectomy by implementing early detection because of recent developments in imaging devices. The AVPTCs are usually associated with large tumors showing ETE or nodal or distant metastasis and higher rates of recurrence and mortality [28]. With the increase in the frequency of early diagnosis, the detection of PTC pathologically belonging to the aggressive variant and other invasive features is also increasing. Therefore, there is growing concern regarding the aggressive treatment of encapsulated AVPTC without invasive features. Limberg et al. recently reported that patients diagnosed with TCV, CCV, and DSV without invasive features had similar overall survival rates as those of patients diagnosed with cPTC [29]. In addition, the histological features of TCV or CCV alone should not be considered to be predictive of unfavorable outcomes [30,31,32]. Therefore, in examining cases of early diagnosis of AVPTC, Andres et al. recommended individualized decision making on the extent of surgery and treatment with adjuvant RAI based on clinical experience [33].

Our study analyzed the difference between HT and BTT in 249 patients diagnosed with AVPTC from 2005 to 2019. Clinicopathologically, more patients who underwent HT were incidentally found to have AVPTC than those who underwent BTT (93.5% vs. 74.9%, *p* = 0.006). This indicated higher cases of early detection through medical check-ups because patients who underwent HT were significantly more likely to have a family history of thyroid cancer or other organ cancers compared with the patients who underwent BTT (26.1% vs. 13.3%, *p* = 0.031). To clarify, early detection owing to regular health check-ups and timely treatments reduce the risk of cancer recurrence. Within an average follow-up duration of 14.9 years, in patients with encapsulated AVPTC with low-to-intermediate-risk factors, HT alone was associated with a recurrence rate of 4.3%. Moreover, there was no significant difference in disease-free survival between the HT and BTT groups. This is similar to the 2–6% recurrence rate in low-risk PTC in the ATA guidelines and lower than the 9% recurrence rate in low-risk differentiated thyroid cancer patients in recently reported systemic reviews [3,34]. In the HT group, two patients with recurrence (4.3%) had T1 tumor limited to one thyroid with microscopic ETE but no gross ETE. Preoperative US suggested the possibility of N1a; however, only HT surgery was performed after consultation with the patients. The maximum size of metastatic LNs was <1 cm, and there were <5 LNs without extranodal extension and LVI. The final pathologic diagnosis was T1N1a stage DSV. Genetic testing showed positive BRAF mutation in one patient and negative in the other patient. The patients underwent high-dose RAI after completion total thyroidectomy with modified radical neck dissection. The patients were treated according to the ATA guidelines, and no cancer-specific mortality occurred during the follow-up period.

Additionally, the optimal tumor size cut-off predictive of recurrence was 1.4 cm, and tumors >1.4 cm and with ETE or locoregional or distant metastasis had a higher risk of recurrence. Meanwhile, the histological variant of cancer did not increase the risk of recurrence. According to the recently revised ATA guidelines, gross ETE, LN ≥ 3 cm, and ≥4 LNs with extranodal extension are listed as high-risk factors; clinical N1, ≥6 LNs, and LVI as intermediate-risk factors; microscopic ETE, <5 LNs; and all LNs < 0.2 cm as low-risk factors of recurrence [3]. However, the results of cross-analysis by risk class within the HT group showed that positivity for multiplicity, clinical N1a, and DSV could help determine the indication for close follow-up after HT or performing a completion total thyroidectomy.

This study has some limitations, such as a small sample size and short follow-up period, because conservative management in the surgical treatment of AVPTC has been drawing attention very recently. In addition, most patients had DSV and SV, and there were fewer cases of TCV, CCV, and HV, which have been reported to show more clinically aggressive features [11,30,31,32,35,36]. Therefore, further long-term prospective and multicenter studies are warranted to evaluate the predictive risk factors and prognosis of AVPTC without aggressive features after lobectomy.

## 5. Conclusions

Close follow-up can be considered without additional surgical treatment in patients with AVPTC that is incidentally found after HT surgery. Moreover, even in at-risk AVPTC patients, it is possible to recommend waiting before performing an immediate completion total thyroidectomy carefully if the patient strongly prefers it. The patients should be provided sufficient consultation with clinicians.

## Figures and Tables

**Figure 1 cancers-14-02757-f001:**
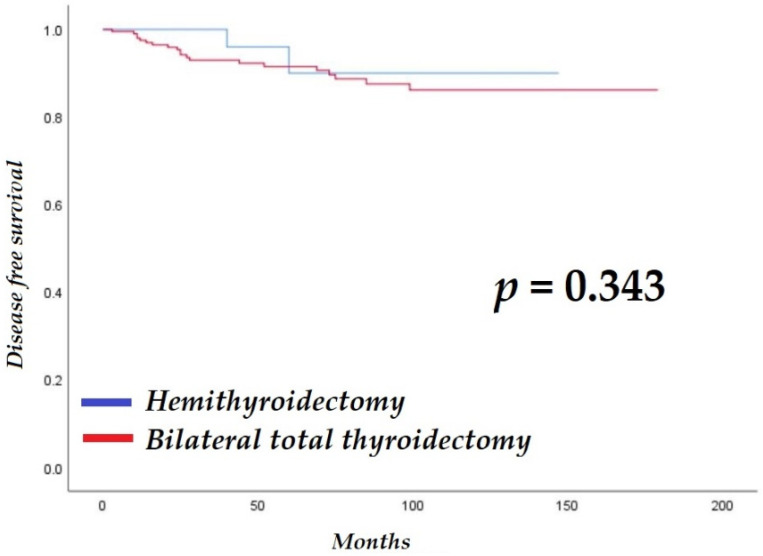
Comparison of disease-free survival between patients with aggressive variants of papillary thyroid cancer who underwent hemithyroidectomy or bilateral total thyroidectomy.

**Table 1 cancers-14-02757-t001:** Comparison of patient characteristics between patients who underwent HT and BTT.

	HT Group (n = 46)	BTT Group (n = 203)	*p*-Value
Sex, male:female	10:36	55:148	0.455
Age (years)	38.0 ± 12.3 (16–66)	35.7 ± 13.1 (7–79)	0.195
Diagnostic Sx			
Incidentaloma	43 (93.5)	152 (74.9)	0.006
Anterior neck mass	3 (6.5)	38 (18.7)	0.044
Lateral neck mass	0 (0.0)	6 (3.0)	0.238
Hoarseness	0 (0.0)	5 (2.5)	0.282
Others	0 (0.0)	2 (1.0)	0.499
Hypothyroidism hx	0 (0.0)	5 (2.5)	0.282
Family hx	12 (26.1)	27 (13.3)	0.031
Thyroid cancer of first degree	7 (15.2)	18 (8.9)	
Thyroid cancer of second degree	0 (0.0)	1 (0.5)	
Others (other organ cancers, hypothyroidism)	5 (10.7)	8 (4.0)	

Data are expressed as number (%) of patients or mean ± standard deviation (range). Statistically significant differences are defined as *p* < 0.05. Abbreviations: HT, hemithyroidectomy; BTT, bilateral total thyroidectomy; Sx, symptoms; and hx, history.

**Table 2 cancers-14-02757-t002:** Comparison of preoperative imaging findings for clinical TNM stage and detection of aggressiveness between patients who underwent HT and BTT.

	HT Group (n = 46)	BTT Group (n = 203)	*p*-Value
Clinical T stage			<0.001
T1	31 (67.4)	47 (23.2)	
T2	9 (19.6)	27 (13.3)	
T3	6 (13.0)	124 (61.1)	
T4	0 (0.0)	5 (2.5)	
Clinical N stage			<0.001
N0	41 (89.1)	45 (22.2)	
N1a	5 (10.9)	22 (10.8)	
N1b	0 (0.0)	136 (67.0)	
Clinical M stage			0.201
M0	46 (100.0)	196 (96.6)	
M1	0 (0.0)	7 (3.4)	
US-guided detection of aggressive tumors	2 (4.3)	76 (37.4)	<0.001

Data are presented as number (%) of patients. Statistically significant differences are defined as *p* < 0.05. Abbreviations: HT, hemithyroidectomy; BTT, bilateral total thyroidectomy; US, ultrasound; and TNM, tumor-node-metastasis.

**Table 3 cancers-14-02757-t003:** Comparison of pathological findings and gene mutation between patients treated with HT and BTT.

	HT Group (n = 46)	BTT Group (n = 203)	*p*-Value
Pathologic cancer subtype			<0.001
Diffuse sclerosing variant	19 (41.3)	163 (80.3)	
Solid variant	19 (41.3)	23 (11.3)	
Tall cell variant	6 (13.0)	13 (6.4)	
Hobnail variant	1 (2.2)	3 (1.5)	
Columnar cell variant	1 (2.2)	1 (0.5)	
Tumor size (cm)	1.2 ± 0.8 (0.4–3.5)	1.9 ± 1.2 (0.3–7.0)	0.226
Tumor number			<0.001
Single	35 (76.1)	90 (44.3)	
Multiple	11 (23.9)	113 (55.7)	
Bilaterality	3 (6.5)	123 (60.0)	<0.001
Variant only	0 (0.0)	73 (36.0)	
Conventional PTC mixed	3 (6.5)	50 (24.6)	
Metastatic LN positive	19 (41.3)	175 (86.2)	<0.001
No. of harvested LNs			
Positive nodes of CCND	1.6 ± 2.4	6.7 ± 5.9	<0.001
Total node of CCND	4.9 ± 2.9	11.1 ± 7.5	0.091
Positive node of MRND		10.2 ± 6.9	
Total node of MRND		48.1 ± 26.5	
Perinodal soft tissue extension positive	1 (2.2)	108 (53.2)	<0.001
Microscopic ETE positive	18 (39.1)	156 (76.8)	<0.001
Gross ETE positive	0 (0.0)	47 (23.2)	<0.001
LVI positive	5 (10.9)	46 (22.7)	0.074
BRAF mutation positive	22 (47.8)	45 (43.3) (n = 104)	0.605
TERT promoter mutation positive	0 (0.0)	4 (16.0) (n = 25)	0.005

Data are expressed as number (%) of patients or as mean ± standard deviation (range). Statistically significant differences are defined as *p* < 0.05. Abbreviations: HT, hemithyroidectomy; BTT, bilateral total thyroidectomy; PTC, papillary thyroid cancer; LN, lymph node; No, number; CCND, central compartment neck dissection; MRND, modified radical neck dissection; ETE, extrathyroidal extension; and LVI, lymphovascular invasion.

**Table 4 cancers-14-02757-t004:** Comparison of postoperative additional treatment and recurrence between patients who underwent HT and BTT.

	HT Group (n = 46)	BTT Group (n = 203)	*p*-Value
Postoperative adjuvant Tx	0 (0.0)	192 (94.6)	<0.001
Low-dose RAI	0 (0.0)	49 (24.1)	
High-dose RAI	0 (0.0)	143 (70.4)	
Recurrence rate	2 (4.3)	22 (10.8)	0.178
Recurrence site			0.397
Local	2 (4.3)	11 (5.4)	
Distant	0 (0.0)	7 (3.4)	
Local and distant	0 (0.0)	4 (2.0)	
Additional Tx after recurrence			0.605
Reoperation	0 (0.0)	2 (1.0)	
High-dose RAI	0 (0.0)	6 (3.0)	
Reoperation with postoperative high-dose RAI	2 (4.3)	11 (5.4)	
Other Tx	0 (0.0)	3 (1.5)	

Data are expressed as number (%) of patients or as mean ± standard deviation (range). Statistically significant differences are defined as *p* < 0.05. Abbreviations: HT, hemithyroidectomy; BTT, bilateral total thyroidectomy; Tx, treatment; and RAI, radioactive iodine.

**Table 5 cancers-14-02757-t005:** Univariate and multivariate Cox regression analysis results for clinical characteristics, clinical image staging, initial pathologic findings, operation type, and adjuvant treatment factors in relation to recurrence of aggressive variants of papillary thyroid cancer.

	Univariate Analysis			Multivariate Analysis		
	HR	95% CI	*p*-Value	HR	95% CI	*p*-Value
Age (years)	1.0	0.9–1.0	0.570			
Female sex	0.5	0.2–1.3	0.155			
Incidentaloma as diagnostic Sx	0.3	0.1–0.7	0.006	0.4	0.2–0.9	0.033
Palpable neck mass as diagnostic Sx	2.9	1.3–6.9	0.013	2.7	1.1–6.4	0.025
FHx positive	1.3	0.4–3.8	0.651			
Clinical T stage						
T1	1.0	Reference	-	1.0	Reference	-
T2	4.7	0.4–51.6	0.207	1.7	0.1–21.3	0.687
T3	9.0	1.2–67.4	0.032	6.6	0.8–53.0	0.075
T4	16.2	1.0–260.2	0.049	5.2	0.3–92.3	0.265
Clinical N stage						
N0	1.0	Reference	-	1.0	Reference	-
N1a	9.3	1.0–89.6	0.053	8.5	0.9–82.4	0.065
N1b	12.7	1.7–94.9	0.013	8.3	1.1–63.4	0.041
US-guided aggressiveness detection	0.8	0.3–2.0	0.679			
Operation type						
HT	1.0	Reference	-			
BTT	2.0	0.5–8.5	0.353			
Subtype of AVPTC						
DSV	1.0	Reference	-			
SV	0.8	0.2–2.8	0.759			
TCV	1.8	0.4–7.7	0.443			
HV	<0.01	N/A	0.988			
CCV	<0.01	N/A	0.989			
Tumor size (cm)	1.4	1.1–1.8	0.006	1.3	1.0–1.7	0.036
Perinodal soft tissue extension positive	1.9	0.9–4.4	0.116			
Microscopic ETE positive	10.2	1.4–75.3	0.023	4.4	0.6–34.2	0.161
Gross ETE positive	4.4	2.0–9.9	<0.001	3.1	1.4–7.0	0.007
Multiplicity positive	1.2	0.5–2.6	0.733			
Bilaterality positive	1.5	0.6–3.3	0.381			
LVI positive	3.0	1.2–7.5	0.019	2.4	0.9–6.6	0.080
Pathologic T stage						
T1	1.0	Reference	-	1.0	Reference	-
T2	2.0	0.5–7.8	0.343	1.2	0.3–5.0	0.800
T3a	5.0	1.3–20.1	0.023	2.7	0.6–12.0	0.185
T3b	5.0	1.5–16.4	0.008	3.4	1.0–11.4	0.045
T4a	8.3	2.8–24.8	<0.001	6.0	1.9–18.8	0.002
Pathologic N stage						
N0	1.0	Reference	-	1.0	Reference	-
N1a	4.6	0.5–38.9	0.167	8.9	0.9–87.3	0.060
N1b	7.4	1.0–55.2	0.052	7.2	0.9–55.9	0.059
Postoperative adjuvant Tx						
No treatment	1.0	Reference	-			
Low-dose RAI	0.5	0.1–3.2	0.491			
High-dose RAI	2.2	0.6–7.4	0.208			

Statistically significant differences were defined as *p* < 0.05. Abbreviations: HR, hazard ratio; CI, confidence interval; Sx, symptom; FHx, family history; AVPTC, aggressive variants of papillary thyroid cancer; DSV, diffuse sclerosing variant; SV, solid variant; TCV, tall cell variant; HV, hobnail variant; CCV, columnar cell variant; HT, hemithyroidectomy; BTT, bilateral total thyroidectomy; N/A, not available; ETE, extrathyroidal extension; LVI, lymphovascular invasion; LN, lymph node; Tx, treatment; RAI, radioactive iodine.

**Table 6 cancers-14-02757-t006:** Univariable Cox proportional hazards regression analysis for tumor size.

	HR	95% CI	*p*-Value	Harrell’s c-Index
Tumor size (continuous)	1.4	1.1–1.8	0.006	0.692
Tumor size (binary)				
<1.4 cm	1.0	Reference	-	0.666
≥1.4 cm	3.3	1.2–9.0	0.016	

Statistically significant differences are defined as *p* < 0.05. Abbreviations: HR, hazard ratio; and CI, confidence interval.

## Data Availability

The data presented in this study are available on reasonable request from Severance Hospital. The data are not publicly available in compliance with the Personal Information Protection Act.

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
