# Peer review of "Predictive Factors Indicative of Hemithyroidectomy and Close Follow-Up versus Bilateral Total Thyroidectomy for Aggressive Variants of Papillary Thyroid Cancer"

_cancers, 2022, doi:10.3390/cancers14112757_

Round 1

Reviewer 1 Report

Lee and colleagues performed a retrospective study exploring the need for additional surgery after lobectomy in patients with aggressive variants of PTC.

Current ATA guidelines suggest that lobectomy can be performed for small unifocal intrathyroidal tumors, while total thyroidectomy with (at least) central neck dissection should be performed for T3 or T4 disease. Basically, the surgical approach is mostly driven by tumor size and local invasiveness rather than histological variant, which is uncommonly diagnosed preoperatively. This is justified by the indolent behavior of encapsulated CCV reported by some authors, but evidence is lacking for the other aggressive variants. Hence, disagreement on the topic exists.

With these premises, the study could be of interest despite the small sample size for this kind of assessment.

Please find below my comments.

Major:

1) The aim of the study, which should be stated clearly at the end of the introduction, is to understand whether the surgical approach (HT vs. BTT) to AVPTC impacts on the outcome of the patient (i.e., recurrence). While there are no significant differences in DFS according to the type of surgery in univariate analysis, great imbalance exists in terms of pathological features of aggressiveness between HT and BTT group. This can explain the higher recurrence rate in the BTT group. Consequently, a multivariate analysis should include the type of surgery along with the pathological features of aggressive disease. Also, the median and interquartile range of follow up would be more useful in the interpretation of the recurrence rate.

2) How is the TNM stage assessed preoperatively? In my opinion, these results could be confounding for the reader. Also, it was reported that preoperative US-guided assessment of lymph node metastases has limited sensitivity. Which is the concordance of the “preoperative” and pathological TNM staging? Please, consider changing the “preoperative TNM” definition and use more reader-friendly definitions, which should be clearly specified in the methods and the results.

3) Page 2, line 67: “the diagnosis was histologically confirmed by US-guided FNA”. This is incorrect. The histological diagnosis can be confirmed only after pathological examination, not by FNA cytology. Please correct. In addition, the US and cytology class, assessed preoperatively, are not reported. It is not clear how HT or BTT could be chosen “according to clinical stage” if US and cytology reports are not present.

4) The ATA risk class is not reported. How well did it perform in predicting recurrence of AVPTC? It would be interesting to compare the actual ATA risk system (which include the upgrade to intermediate risk for aggressive variants) with the ATA risk based on clinical-pathological features of risk, without the upgrading of aggressive variants PTC which are intrathyroidal tumors with less than 5 LN metastases smaller than 0.2 cm. This could help understand whether incidental intrathyroidal AVPTC may be considered low risk tumors.

Minor:

5) It would be interesting to provide the BRAF and TERT promoter mutation rate per variant.

6) In the methods, serum markers are mentioned, but no results are reported. Were serum markers different among groups? Were they useful in predicting recurrence?

7) The references are not completely up to date. Recent large monocentric series or metanalyses are present in literature, especially for TCV and DSV.

8) In Tables 5 and 6, it is not clear how some features were managed in the Cox regression. For instance, were age and tumor size used as continuous variables? Which is the reference class for sex?

9) In the discussion, lines 328-341 are actually results and should be moved in the appropriate section.

10) line 323 “histological characteristics of cancer did not increase the risk of recurrence”. Maybe the authors referred to histological variant, since pathological features did increase the risk of recurrence as shown in Table 6.

Author Response

Dear reviewer,

Thank you for your careful review.

Based on the review, our research seems to have become more concrete and clear.

1) a multivariate analysis should include the type of surgery along with the pathological features of aggressive disease.

-> There was a misunderstanding in the contents of the ticket about what you said. After consulting with the statistics team, we marked both univariate and multivariate analysis results to make the table easier to understand, and integrated it into Table 5 to create a new table. As a result, the scope of the surgery did not obtain significant results from the univariate analysis, so the multivariate analysis was not performed.

1-2) median and interquartile range of follow up would be more useful in the interpretation of the recurrence rate.

-> As you recommended, we added the median and interquartile range of follow up between two groups on line 216; “The average, median, and interquartile ranges of follow-up for the HT and BTT groups were 55.1±39.0 months (range, 12–147 months), 42.5 months, and 54 months, and 77.0±44.9 months (range, 11–179 months), 73.0 months, and 79 months, respectively.”

2) Please, consider changing the “preoperative TNM” definition and use more reader-friendly definitions, which should be clearly specified in the methods and the results.

-> As you said, preoperative TNM stage could confuse readers. Therefore, we changed all “preoperative” to “clinical” TNM stage.

3) Page 2, line 67: “the diagnosis was histologically confirmed by US-guided FNA”. This is incorrect. The histological diagnosis can be confirmed only after pathological examination, not by FNA cytology. Please correct.

-> As you said, "histologically" has been deleted, and the content has been modified to "the diagnosis of cancer was confirmed by US-guided fine-needle aspiration." on line 70.

3-2) In addition, the US and cytology class, assessed preoperatively, are not reported. It is not clear how HT or BTT could be chosen “according to clinical stage” if US and cytology reports are not present.

-> We admit that there was a lack of further explanations for reporting system. So, we added " Clinical classification of cancer and lymph nodes was performed according to the Korean Thyroid Imaging Reporting and Data System by radiologists at our institution. " to line 71 with reference and deleted the phrase "according to clinical stage".

4) The ATA risk class is not reported. How well did it perform in predicting recurrence of AVPTC? It would be interesting to compare the actual ATA risk system (which include the upgrade to intermediate risk for aggressive variants) with the ATA risk based on clinical-pathological features of risk, without the upgrading of aggressive variants PTC which are intrathyroidal tumors with less than 5 LN metastases smaller than 0.2 cm. This could help understand whether incidental intrathyroidal AVPTC may be considered low risk tumors.

-> I added the following information about what you said and explained a little more specifically about the risk class of the groups that was proceeded with sub-analysis on line 252 and 319.

“Patients in the HT group had none or one to two of risk factors, including microscopic ETE, LVI, ≥4 LNs with extranodal extension, and ≥6 LNs. The patients were further divided into two groups: with no/low-risk factor and with intermediate or high-risk factor according to revised ATA guidelines [3]..”

“According to the recently revised ATA guidelines, gross ETE, LN ≥3 cm, and ≥4 LNs with extranodal extension are listed as high-risk factors; clinical N1, ≥6 LNs, and LVI as intermediate-risk factors; and microscopic ETE, <5 LNs; and all LNs <0.2 cm as low-risk factors of recurrence [3].”

5) It would be interesting to provide the BRAF and TERT promoter mutation rate per variant.

-> We added “The BRAF and TERT mutation rates were 38.9% and 8.1% for DSV, 36.4% and 0.0% for SV, 94.1% and 9.1% for TCV, 50% for HV, and 0.0% and 0.0% for CCV, respectively.” to line 174.

6)  In the methods, serum markers are mentioned, but no results are reported. Were serum markers different among groups? Were they useful in predicting recurrence?

-> We are routinely testing serum marker, but we did not include it in the content because no significant usefulness was found between the two groups in AVPTC patients. If you think the content is unnecessary, I will delete it.

7) The references are not completely up to date. Recent large monocentric series or metanalyses are present in literature, especially for TCV and DSV.

-> I added the latest papers as a reference to the introduction and discussion.

8) In Tables 5 and 6, it is not clear how some features were managed in the Cox regression. For instance, were age and tumor size used as continuous variables? Which is the reference class for sex?

-> Sex was revised to female sex, and age and tumor size were used as continuous variables. Additionally, like we said before, we agree that Table 5 and 6 are confusing the analysis results. We revised the table again with the advice of a statistician, set Table 5 and 6 as Table 5, and revised the interpretation of the results.

9) In the discussion, lines 328-341 are actually results and should be moved in the appropriate section.

-> The results on subanalysis moved to the back of Table 5 in result, as mentioned earlier. As a result, the content about discussion has been modified.

10) line 323 “histological characteristics of cancer did not increase the risk of recurrence”. Maybe the authors referred to histological variant, since pathological features did increase the risk of recurrence as shown in Table 6.

-> I'm sorry for the misunderstanding. The sentence was amended to " histological variant of cancer.".

I attached the revised manuscript because I modified it a lot.

Please feel free to contact me if I misunderstood or if you have any further questions.

Thank you again for giving us your valuable time. 

Reviewer 2 Report

Line 19: Is it multicenter or one hospital?

Line 22: touching means palpable tumor?

Line 23: wide CI indicate imprecision for N1b and TNM. Also both are correlated, so should not be included together in the regression model.

Line 27: "Additional surgery" is unclear what categories. "Completing total thyroidectomy" might be more comprehensive.

Line 46-47: statement need to be rephrased to be more clear.

Line 49: "a" need to be removed, because patients could have other types of BRAF mutation rather than V600E.

Line 82: Definition of recurrence

Line 119: Please add type of tests in footer of tables.

Lines 126=127: each % could be after the category immediately.

Is there data for surgical margin postoperative?

Table 5: TC is the only cancer staging which includes age, so it will be biased when included in the cox model. What is the reference for incidentomal and Ant neck mass.

Table 6: What s tumor size cutoff that increase the risk? Model assessment will show that some covariates have colinearity, and thus, could be emplyed as univariate analysis then later on omit on multivariate.

Suggest estimating risk score from cox regression model and assess its prognostic accurate in ROC curve, nomogram, calibration plot.

Conclusion:

The title of the article indicates as if same patients will undergo type of surgery, then introduced to more therapeutic procedure. therefore, it was misleading. Suggest modifying the tone of the theme within the manuscript because it does not fit study design.

Author Response

Thank you for your careful review.

Based on the review, our research seems to have become more concrete and clear.

  1. Line 19: Is it multicenter or one hospital?

-> I added “in a single institution.” on line 19.

  1. Line 22: touching means palpable tumor?

-> Yes, it is. I modified the contents to a palpable tumor.

  1. Line 23: wide CI indicate imprecision for N1b and TNM. Also both are correlated, so should not be included together in the regression model.

-> we agree that the results of multivariate analysis are not clear, and table 5 and 6 are confusing the analysis results. We revised the table with the advice of a statistician, set Table 5 and 6 as Table 5, and modified the explanation in the result of abstract and the 3.3 result section. In addition, as you recommended, TNM stage, which can be bias in the multivariate analysis, was excluded from the analysis.

  1. Line 27: "Additional surgery" is unclear what categories. "Completing total thyroidectomy" might be more comprehensive.

-> As you said, we revised “additional surgery” to “completion total thyroidectomy".

  1. Line 46-47: statement need to be rephrased to be more clear.

-> I switched to “conversely” instead of “Meanwhile” because this sentence means that SV, unlike DSV, had a rather poor prognosis.

  1. Line 49: "a" need to be removed, because patients could have other types of BRAF mutation rather than V600E.

-> I'm sorry to make you misunderstand. I deleted "a" from the sentence you said.

  1. Line 82: Definition of recurrence

-> We added “For patients showing signs of recurrence on postoperative imaging, FNA was used to confirm the recurrence.” on line 83 as the definition of recurrence.

  1. Line 119: Please add type of tests in footer of tables.

-> I added this in footer of tables.

  1. Lines 126-127: each % could be after the category immediately.

-> I revised this sentence to “Most patients in the HT group had a T1 stage (67.4%) and N0 stage (89.1%) disease. Meanwhile, the BTT group had significantly more advanced T and N stage compared with the HT group (P<0.001 and P<0.001, respectively).” on line 131.

  1. Is there data for surgical margin postoperative?

-> We routinely checked the surgical margin during surgery and performed further resection if necessary, and all the surgeries of hemithyroidectomy were margin negative and there was no information on the length.

  1. Table 5: TC is the only cancer staging which includes age, so it will be biased when included in the cox model. What is the reference for incidentomal and Ant neck mass.

-> We agree that TC is cancer that affects age. However, the age distribution of TC patients in our data was 18-77 years old, and when analyzed with the Mann-Whitney U test between TC and except TC, it was confirmed that there was no significant difference as P = 0.395. Therefore, in Cox analysis, statistics were performed together without excluding TC.

In addition, palpable mass defined it as a case of direct appeal from a patient, and incidentaloma defined it as a case in which a patient does not have any specific symptoms.

  1. Table 6: What s tumor size cutoff that increase the risk? Model assessment will show that some covariates have colinearity, and thus, could be emplyed as univariate analysis then later on omit on multivariate. Suggest estimating risk score from cox regression model and assess its prognostic accurate in ROC curve, nomogram, calibration plot.

-> As you advised, I used Contal & O’quigley’s method and Cox proportional hazards regression analysis to get an optimal tumor size cut-off using R package. This showed HR 3.3 and 95% CI 1.2-9.0 when tumor size was over 1.4 cm, and Harrell's c-index was 0.666. I added this content by making new table 6.

It is good to conduct additional tests for prognostic accurate, but it is difficult to generalize in clinical trials because it is not well reproduced with data-based results, and there is no validation set that can verify cut-off point, so it could be supplemented through multicenter research later.

  1. Conclusion:

The title of the article indicates as if same patients will undergo type of surgery, then introduced to more therapeutic procedure. therefore, it was misleading. Suggest modifying the tone of the theme within the manuscript because it does not fit study design.

->We revised our title to "Predictive Factors Indicative of Hemithyroidectomy and Close Follow-up Versus Bilateral Total Thyroidectomy for Aggressive Variants of Papillary Thyroid Cancer."

I attached the revised manuscript because I modified it a lot.

Please feel free to contact me if I misunderstood or if you have any further questions.

Thank you again for giving us your valuable time. 

Round 2

Reviewer 1 Report

The manuscript has been improved.

However, I have further comments:

1) “TERT mutation” should be “TERT promoter mutation”;

2) line 70-71: “the diagnosis of cancer was confirmed by US-guided fine-needle aspiration (FNA).” As I said in the first round of review, the diagnosis of cancer can be confirmed only at pathological examination. FNA cytology can help classify a nodule as benign or malignant preoperatively, but the confirmation can be done at histology. In addition, it would be interesting to report the cytology and US class of the nodules, and compare them between the HT and BTT group.

3) Altough the findings are interesting, the small sample size and the short follow up are important limitations for this kind of assessment. Please, consider to soften the conclusion.

Author Response

Dear reviewer, 

Your advice has refined a lot of my manuscript.

I have revised the following for additional comments.

1) “TERT mutation” should be “TERT promoter mutation”;

->Thank you for the delicate correction. I modified five "TERT mutations" to "TERT promoter mutation."

2) line 70-71: “the diagnosis of cancer was confirmed by US-guided fine-needle aspiration (FNA).” As I said in the first round of review, the diagnosis of cancer can be confirmed only at pathological examination. FNA cytology can help classify a nodule as benign or malignant preoperatively, but the confirmation can be done at histology. In addition, it would be interesting to report the cytology and US class of the nodules, and compare them between the HT and BTT group.

-> As you said, I modified “diagnosis of cancer” to “preoperative malignancy classification” on line 70. In addition, the US class of the nodules you mentioned has content on table 2, so I added the following information about the results of FNA cytology of the nodule on line 137.

“The FNA cytology results for the nodule were 2 (4.3%), 1 (2.2%), 5 (10.9%), 21 (45.7%), and 17 (37.0%) for nondiagnostic, benign, atypia/follicular lesions of undetermined significance, suspicious for malignancy, and malignant, respectively, in the HT group. However, in the BTT group, there were only 15 (7.4%), 142 (70.0%), and 46 (22.7%) for atypia/follicular lesions of undetermined significance, suspicious for malignancy, and malignant, respectively, by significant difference (P <0.001).”

3) Altough the findings are interesting, the small sample size and the short follow up are important limitations for this kind of assessment. Please, consider to soften the conclusion.

-> I agree with what you are worried about. Therefore, the contents of the conclusion were changed as follows.

“Close follow-up can be considered to performed without additional surgical treatment in patients with AVPTC that is incidentally found after HT surgery. Moreover, even in at-risk AVPTC patients, it is possible to recommended to waiting before per-forming an immediate completion total thyroidectomy carefully if the patient strongly prefers it. The patients should be provided sufficient consultation with clinicians.”

Thank you again for your careful review.

Round 3

Reviewer 1 Report

I have no further comments.